# Optimizing Neuro-Oncology Imaging: A Review of Deep Learning Approaches for Glioma Imaging

**DOI:** 10.3390/cancers11060829

**Published:** 2019-06-14

**Authors:** Madeleine M. Shaver, Paul A. Kohanteb, Catherine Chiou, Michelle D. Bardis, Chanon Chantaduly, Daniela Bota, Christopher G. Filippi, Brent Weinberg, Jack Grinband, Daniel S. Chow, Peter D. Chang

**Affiliations:** 1Department of Radiology, University of California, Irvine, Orange, CA 92868, USA; mshaver@uci.edu (M.M.S.); pkohante@uci.edu (P.A.K.); clchiou@uci.edu (C.C.); mbardis@uci.edu (M.D.B.); cchantad@uci.edu (C.C.); changp6@uci.edu (P.D.C.); 2Department of Neurology, University of California, Irvine, Orange, CA 92868, USA; dbota@uci.edu; 3Department of Radiology, North Shore University Hospital, Manhasset, NY 11030, USA; sairaallapeikko@gmail.com; 4Department of Radiology, Emory University School of Medicine, Atlanta, GA 30322, USA; Brent.d.weinberg@gmail.com; 5Department of Radiology, Columbia University, New York, NY 10032, USA; jackgrinband@gmail.com

**Keywords:** glioma, glioblastoma, machine learning, artificial intelligence, deep learning, neural network

## Abstract

Radiographic assessment with magnetic resonance imaging (MRI) is widely used to characterize gliomas, which represent 80% of all primary malignant brain tumors. Unfortunately, glioma biology is marked by heterogeneous angiogenesis, cellular proliferation, cellular invasion, and apoptosis. This translates into varying degrees of enhancement, edema, and necrosis, making reliable imaging assessment challenging. Deep learning, a subset of machine learning artificial intelligence, has gained traction as a method, which has seen effective employment in solving image-based problems, including those in medical imaging. This review seeks to summarize current deep learning applications used in the field of glioma detection and outcome prediction and will focus on (1) pre- and post-operative tumor segmentation, (2) genetic characterization of tissue, and (3) prognostication. We demonstrate that deep learning methods of segmenting, characterizing, grading, and predicting survival in gliomas are promising opportunities that may enhance both research and clinical activities.

## 1. Introduction

Radiographic assessment with magnetic resonance imaging (MRI) is widely used to characterize gliomas, which represent a third of all brain tumors and 80% of all primary malignant brain tumors [1]. From a clinical perspective, imaging is often used preoperatively for diagnosis and prognostication, and post-operatively for surveillance. From a research perspective, MRI assessment provides a standardized method with which to establish patient baselines and identify endpoints for monitoring response to therapies for patient clinical trial enrollment and participation. Unfortunately, obtaining reliable, quantitative imaging assessment is complicated by the variegation of glioma biology, which is marked by heterogeneous angiogenesis, cellular proliferation, cellular invasion, and apoptosis [2].

Several approaches have emerged to standardize visual interpretation of malignant gliomas for tissue classification. For example, the Visually AcceSAble Rembrandt Images (VASARI) feature set is a rules-based lexicon to improve reproducibility of glioma interpretation [3]. Gutman et al. [4] successfully implemented this analysis for routine structural images within 75 glioblastoma multiforme (GBM) patients from The Cancer Genome Atlas (TCGA) portal and observed lower levels of contrast enhancement within proneural GBMs and lower levels of the non-enhanced tumor for mesenchymal GBMs. Human-designed rule-based systems such as VASARI have improved the reproducibility of glioma interpretation [3,5,6], but a small number of numeric descriptors are inadequate to capture the complexity of a typical MRI scan with over a million voxels, constituting a “Big Data” problem.

Machine learning approaches are uniquely suited to tackle such Big Data challenges. They utilize computational algorithms to parse and learn from data, and ultimately make a determination given the input variables. Machine learning has been used to train computers for pattern recognition, a task that usually requires human intelligence (Figure 1) [7]. Classic machine learning approaches employ human-designed feature extraction to distinguish tumor characteristics, and this has improved the accuracy of identifying tumor features on imaging [8]. For example, Hu et al. [9] utilized hand-crafted features derived from textural metrics to first characterize 48 biopsies from 13 patients, which were ultimately used as input to a decision tree classifier to predict underlying tumor molecular alterations. In Hu et al.’s example, the textural analysis takes advantage of manually identified features reflecting *a priori* (pre-selected) human expert assumptions about imaging metrics relevant to tumor biology [9]. While this exploratory study provided a framework for future studies evaluating image-based signatures of the intra-tumoral variability, the performance of this classic machine learning approach is limited to the ability of the *a priori* features alone to capture all the relevant image variance needed to predict tumor mutations [10].

By contrast, deep learning approaches do not require pre-selection of features and can, instead, learn which features are most relevant for classification and/or prediction. Deep learning, a subset of machine learning, can extract features, analyze patterns, and classify information by learning multiple levels of lower and higher-order features [11]. Lower-order features, for example, would include corners, edges, and other basic shapes. Higher-order features would include different gradations of image texture, more refined shapes, and image-specific patterns [12]. Furthermore, deep learning neural networks are capable of determining more abstract and higher-order relationships between features and data [11]. The state-of-the-art approach for image classification is currently deep learning through convolutional neural networks (CNNs). For reference, CNN approaches represent all recent winning entries within the annual ImageNet Classification Challenge, consisting of over one million photographs in 1000 object categories, with a 3.6% classification error rate to date [13,14]. CNN approaches model the animal visual cortex by applying a feed-forward artificial neural network to simulate multiple layers of inputs where the individual neurons are organized to respond to overlapping regions within a visual field [15]. Because it functions similarly to a human brain in recognizing, processing, and analyzing visual image features, a CNN approach is very effective for image recognition applications [16].

The purpose of this review is to summarize deep learning applications used in the field of glioma detection, characterization, and outcome prediction, with a focus on methods that seek to (1) quantify disease burden, (2) determine textural and genetic characterization of tumor and surrounding tissue, and (3) predict prognosis from imaging information. 

## 2. Pre and Post-Operative Tumor Segmentation: Quantification of Disease Burden

Quantitative metrics are needed for therapy guidance, risk stratification, and outcome prognostication, both pre and post-operatively [17]. Simple radiographic monitoring with freehand measurements of the amount of contrast-enhancing tumor in 2 or 3 planes is commonly used for estimating disease burden; however, single-dimensional techniques may be inaccurate, and not reflective of change in actual tumor burden [18], particularly given the propensity of high-grade tumors to grow in a non-uniform and unpredictable fashion. Manual brain tumor segmentation represents a potential solution and involves separating tumor tissues such as edema and necrosis from normal brain tissue such as gray matter, white matter, and cerebrospinal fluid [19]. However, manual segmentation is both time consuming and is subject to reader variability, making fast and reproducible segmentations challenging. Machine learning approaches represent a potential solution to meet these challenges. A summary of recent machine learning architectures and approaches used to segment both pre and post-operative GBMs is listed in Table 1. 

The Multimodal Brain Tumor Image Segmentation (BraTS) dataset, which was created in 2012, has been extensively used to demonstrate the efficacy of deep learning applications in segmenting pre-operative GBMs [25]. The BraTS dataset gave deep learning designers and programmers access to hundreds of GBM images and has become a benchmark for GBM segmentation performance [26]. For example, using the 2017 BraTS data [27], Chen et al. [21] developed a neural network which hierarchically segmented the necrotic and non-enhancing tumor, peritumoral edema, and enhancing tumor, resulting in mean Dice coefficients for whole tumor, enhancing tumor, and core of 0.72, 0.81, and 0.83, respectively. The Dice coefficient is a statistic used for comparing the spatial overlap of two binary images and is routinely used for tissue classification assessment [20]. It ranges between 0 and 1, where 0 indicates no overlap and 1 indicates exact overlap. In contrast to Chen et al., Havaei’s study [22] exploits both local features and global contextual features simultaneously by using a CNN that can extract small image details and the context of the image. Mean Dice coefficients of 0.81, 0.58, and 0.72 for whole tumor, enhancing tumor, and core were achieved. More importantly, every winning entry for the BraTS competition has since been a neural network implementation.

In post-operative surveillance, MRI imaging with gadolinium contrast is the standard for determining tumor growth (denoted by increased contrast-enhanced growth) and tumor response (denoted with decreases in CE tumor size) [3,28]. Both Response Assessment for Neuro-Oncology (RANO) and Macdonald criteria for GBM assessment rely on two-dimensional measurements of the contrast-enhanced area, the product of the two maximum diameters of the enhancing tissue [3]. Although this approach is simple, widely available, and requires minimal training, there are concerns about its reproducibility and accuracy. For example, while linear measurements may be sufficient for rounded nodular lesions, these measurements may be inaccurate for tumors with irregular margins, a feature common in high-grade gliomas given their propensity for necrosis and eccentric growth [11]. The variegation of glioma pathogenesis and subsequent response to therapy contributes to myriad patterns observed on imaging and may result in classifying effective treatments as ineffective or ineffective treatments as effective, underscoring the need for reliable, reproducible, and accurate tools of surveillance [29]. Volumetric assessment techniques have shown better accuracy in determining tumor size when compared to linear methods in several studies [30,31,32]. Additionally, Dempsey et al. [33] demonstrated that volumetric analysis of tumor size served as a stronger predictor of survival compared to linear-based techniques. Kanaly et al. [34] demonstrated that a semi-automatic approach to brain tumor volume assessment reduced inter-observer variability while being highly reproducible. To obtain more accurate approximations of tumor size, quantifying the entire tumor in three dimensions would provide more precise measurements (Figure 2) [35].

For accurate quantification, deep learning has been applied to estimate tumor volume in post-operative GBMs. Yi et al. [23] used the 2015 BraTS dataset, which contained post-operative GBMs [26] and implemented a CNN that combined 4 imaging modalities (T1 pre-contrast, T1 post-contrast, T2, and FLAIR) at the beginning of the CNN architecture. Yi et al. also used special detection of the tumor edges for faster training and the performance was 0.89, 0.80, and 0.76 for whole tumor, enhancing tumor, and core. Rao et al. [24] examined post-operative GBMs by training with the 2015 BraTS dataset and applied a deep learning network combined with a random forest for separately classifying non-tumor tissue, necrosis edema, non-enhancing tumor, and enhancing tumoral tissue. These studies demonstrated the capabilities of deep learning approaches to improve quantification of disease burden.

## 3. Characterization: Pseudoprogression

Distinguishing pseudoprogression from true tumor progression is important for identifying appropriate treatment options for glioma patients. Presently, the only accepted methods to distinguish true progression of disease (PD) from treatment-related pseudoprogression of disease (psPD) is invasive tissue sampling and short interval clinical follow-up with imaging, which may delay and compromise disease management in an aggressive tumor [36,37]. RANO criteria includes methods for psPD evaluation but remains limited [38]. For example, Nasseri et al. [39] demonstrated that many psPD cases were not accurately identifiable with RANO criteria. In addition, Abbasi et al.’s [40] meta-analysis demonstrated that cases of psPD in high-grade gliomas were more frequent than most studies reported, largely comprising 36% of all post-treatment MRI contrast enhancement-identified cases. 

Classic machine learning methods have been more robustly explored than deep learning models for characterizing psPD on imaging. Hu et al.’s [41] Support Vector Machine (SVM) approach (a traditional linear machine learning technique) examining multiparametric MRI data from 31 patients yielded an optimized classifier for psPD with a sensitivity of 89.9%, specificity of 93.7%, and area under ROC curve of 0.944. Other machine learning methods explored in assessing psPD have included unsupervised clustering [42] and spatio-temporal discriminative dictionary learning schemes [43]. Though deep learning methods have been less often used, they are showing promise for characterizing psPD versus true PD. Jang et al. [44] assessed a hybrid approach that coupled a deep learning CNN algorithm to a classical machine learning, long short-term memory (CNN-LSTM) method to determine psPD versus tumor PD in GBM. Their dataset consisted of clinical and MRI data from two institutions, with 59 patients in the training cohort and 19 patients in the testing cohort. Their CNN-LSTM structure, utilizing both clinical and MRI data, outperformed the two comparison models of CNN-LSTM with MRI data alone and a Random Forest structure with clinical data alone, yielding an AUC (area under the curve) of 0.83, an AUPRC (area under the precision-recall curve) of 0.87, and an F-1 score of 0.74 [44]. This example indicates that utilization of a deep learning approach can outperform a more traditional machine learning approach in analyzing images.

## 4. Characterization: Radiogenomics

Characterizing genetic features of gliomas is important for both prognosis and predicting response to therapy. For example, isocitrate dehydrogenase (IDH)-mutant GBMs are characterized by significantly improved survival than IDH-wild GBMs (31 months vs. 15 months) [45,46]. Recognition of the importance of genetic information has led the World Health Organization (WHO) to place considerable emphasis on the integration of genetic information, including IDH status, into its classification schemes in its 2016 update [47]. Regarding treatment response, it is becoming increasingly evident that GBMs’ differing genetic attributes also result in mixed responses [48]. One of the early mutations discovered was O^6^-methylguanine-DNA methyltransferase (MGMT) promoter silencing, which reduces tumor cells’ ability to repair DNA damage from alkylating agents such as temozolomide (TMZ). Hegiel et al. [49] subsequently observed that MGMT promoter methylation silencing was observed in 45% of GBM patients and demonstrated a survival benefit when treated with a combination of TMZ and radiotherapy versus radiotherapy alone (21.7 months versus 15.3 months). 

Currently, genomic profiling is performed on tissue samples from enhancing tumoral components. However, securing tumor-rich biopsies is challenging and a TCGA report observed that only 35% of submitted biopsy samples contained adequate tumoral content [50]. In addition, tumors may be surgically inaccessible when eloquent areas of the brain are involved. Furthermore, biospecimens are typically required for clinical trial entry, which may be delayed as patients wait weeks before and after resection for genetic results to return. The growing field of radiomics—the extraction and detection of quantitative features from radiographic imaging through computerized algorithms—seeks to address this problem by extracting quantitative imaging features that may lead to a better understanding of the characteristics of a particular disease state [51,52,53]. Imaging features of gliomas have been linked to genetic features [54] and are strongly correlated with particular subtypes of glioma and overall patient survival [4,28]. This has led to the creation of tools and methods such as VASARI, which seek to standardize glioma characterization through identification of outlined imaging features [3]. Radiogenomics, a branch of radiomics, holds particular promise as a non-invasive means of determining tumor genomics through non-labeled radiographic imaging. This push towards utilizing MR imaging to classify and characterize gliomas has made deep learning applications an especially appealing means of quickly and automatically characterizing gliomas.

Levner et al. [55] was one of the earliest groups to use neural networks to predict tumoral genetic subtypes from imaging features. Their model sought to predict MGMT promoter methylation status in newly diagnosed GBM patients using features extracted by space-frequency texture analysis based on the S-transform of brain MRIs. Levner’s group achieved an accuracy of 87.7% across 59 patients, among which 31 patients had biopsy-confirmed MGMT promoter methylated tumors [55]. Korfiatis et al. [56] compared 3 different Residual CNN methods to predict MGMT promoter methylation status on 155 brain MRIs without a distinct tumor segmentation step. Residual CNNs employ many more layers than the traditional CNN architectures for training data [13,56,57]. It was shown that the ResNet50 layer architecture outperformed ResNet34 and ResNet18, achieving high accuracies of 94.90%, 80.72%, and 76.75%, respectively, despite the absence of a tumor segmentation step [56]. Ken Chang et al. [58] applied a Residual CNN to 406 preoperative brain MRIs (T1 pre- and post-contrast, T2, and FLAIR) acquired from 3 different institutions ranging from Grade II to Grade IV gliomas to predict IDH mutation status. Their group achieved IDH prediction accuracy of 82.8% for the training set, 83.0% for the validation set, and 85.7% for the testing set. They noted slight increases of each when patient age at diagnosis was included [58]. 

Many deep learning approaches have also managed to successfully characterize single tumoral genetic mutations on brain imaging. Chang et al. [59] described a CNN for accurate prediction of IDH1, MGMT methylation, and 1p/19q co-deletion status from 256 brain MRIs from the Cancer Imaging Archives Dataset. They achieved an accuracy of 94% for IDH status, 92% for 1p/19q co-deletion status, and 83% for MGMT promoter methylation status. In addition, they applied a dimensionality reduction approach (principal component analysis) to the final CNN layer to visually display the highest ranking features for each category, an important step towards explainable deep learning approaches (Figure 3) [59]. 

A variety of other deep learning methods have been employed to assess genetic and cellular character on radiographic imaging. Several other studies in addition to those discussed have been listed in Table 2 [55,60,61,62,63,64,65,66]. With the advent of these new technologies, radiogenomic characterization of tumors continues to grow as a means of non-invasive assessment of brain tumors.

## 5. Prognostication 

Machine learning approaches are increasingly being explored as methods of automatically and accurately grading and predicting prognosis in glioma patients. Various machine learning approaches, including Support Vector Machine (SVM) classifiers, have been utilized in grading and evaluating the prognosis of gliomas. These strategies predominantly rely on extracting distinguishing features of gliomas from pre-existing patient data to build a prototype model. Both of these tasks portend important clinical considerations such as monitoring disease progression and recurrence, developing personalized surgical and chemo-radiotherapy treatment plans, and examining treatment response.

For example, Zhang et al. [67] explored the use of SVM in grading gliomas in 120 patients. These investigators combined SVM with the Synthetic Minority Over-sampling Technique (i.e., over-sampling the abnormal class and under-sampling the normal class) and were able to classify low-grade and high-grade gliomas with 94–96% accuracy. More recently, SVM classifiers have been applied in evaluating glioma patients’ prognosis. In a study with 105 high-grade glioma patients, Macyszyn et al. [68] demonstrated that their SVM model could classify patients’ survival into short or long-term categories with an accuracy range of 82–88%. In another study with 235 patients, Emblem et al. [69] developed a SVM classifier utilizing histogram data of whole tumor relative cerebral blood volume (rCBV) to predict pre-operative glioma patient overall survival (OS). The sensitivity, specificity, and AUC were 78%, 81%, and 0.79 at 6 months and 85%, 86%, and 0.85 at 3 years for OS prediction [69]. A systemic literature review by Sarkiss et al. [70] provides evidence of the many applications of machine learning in exploring this topic. Twenty-nine studies from 2000 to 2018 totaling 5346 patients and using machine learning in neuro-oncology were included, and for 2483 patients with prediction outcomes showed a sensitivity range of 78–93% and specificity range of 76–95% [70]. These studies highlight the efficacy and accuracy of machine learning-based models in determining patients’ OS compared to human readers (board-certified radiologists), which are prone to variability and subjectivity inherent in human perception and interpretation.

Deep learning-based radiomics models have also been proposed for survival prediction of glioma patients. For example, Nie et al. [71] hybridized a traditional SVM approach with a deep learning architecture. This deep learning architecture involved a three-dimensional CNN that extracted defining features from pre-existing brain tumors. When combined with SVM, this two-step method achieved an accuracy of 89% in predicting OS in a cohort of 69 patients with high-grade gliomas. Their findings suggest that deep learning methods coupled with linear machine learning classifiers can result in the accurate prediction of OS.

While still in their introductory stages, deep learning approaches are promising tools for accurate and expeditious interpretation of complex data that minimize human error and bias.

## 6. Challenges

It is important to recognize that several substantive challenges for deep learning radiographic analysis include the relative lack of annotated data [72] and limited algorithm generalizability, as well as barriers to integration into the clinical workflow. Proper training and convergence of deep learning algorithms currently requires a tremendous volume of high-quality, well-annotated data; however, such datasets are difficult to aggregate in large part due to regulatory bottlenecks that limit sharing of patient data between institutions. In addition, existing analysis of retrospective data acquired during routine clinical care may be inconsistent (e.g., scanned at variable time intervals and/or follow-up). Finally, even if large cohorts across many hospitals can be aggregated, annotation is a time-consuming process requiring a high level of expertise. Given that manual annotations are often time-consuming, future development of customized semi-automated labeling tools and iterative re-annotation strategies may provide an effective solution by relying on machine learning techniques to provide initial ground-truth estimates, which are then refined by human experts [15]. This may allow for faster, better-quality annotated input data necessary to effectively train deep learning algorithms.

Related to the problem of small datasets is the well-documented capacity of a deep learning algorithm with millions of parameters to over-fit to a single, specific training cohort, resulting in an artificially inflated algorithm accuracy [73]. This is especially true given the relatively limited curated datasets currently observed in radiographic research. In addition to striving for large, heterogeneous datasets, there have been several methods developed that attempt to address this limitation, including the addition of feature dropout described by Srivastava et al. [73], L2 regularization [74], and batch normalization [75].

Additional challenges relate to the deployment of deep learning applications in a clinical setting. Challenges to employing computer-aided diagnosis (CAD) within the radiology workflow mirror expected challenges to the integration of AI methods. Radiologists are likely to reject newer technologies that prove disruptive to their workflow or have interfaces which are difficult to access and are not readily available on a normal PACS viewer [76,77]. A study by Karssemeijer et al. [78] examining the efficacy of CAD in breast mass detection on mammogram demonstrated overall lower performance of the CAD system compared with expert readers due to the larger number of false positives recorded for CAD. To be clinically useful, CAD (and by extension, deep learning) approaches would need to provide improved diagnostic capabilities while also optimizing normal workflow [79]. FDA regulatory restrictions also constrain the deployment of deep learning tools into a clinician’s toolbox. Currently there are few acceptable regulatory pathways for approval of deep learning for clinical use. One attempt to address this by the FDA is through the creation of the Digital Health Software Precertification (Pre-Cert) program [80]. More recently, the FDA released a proposed regulatory framework that seeks to allow for oversight of machine learning and other continuous learning models as medical devices [81]. While steps are underway to address the regulatory gap between deep learning research and clinical utility, this still exists as a barrier to effective clinical deployment.

## 7. Conclusions

Deep learning applications continue to provide effective solutions to problems in medical image analysis. Radiological sciences and the growing field of radiomics are well poised to incorporate deep learning techniques well-suited to quick image analysis. There are many opportunities rife for exploration in deep learning analysis of gliomas on radiographic imaging, including determining tumor heterogeneity, more extensive identification of tumor genotype, cases of progression and pseudoprogression, tumor grading, and survival prediction. This type of imaging analysis naturally bolsters the possibility for precision medicine initiatives in glioma treatment and management. Deep learning methods for segmenting, characterizing, grading, and predicting survival in glioma patients have helped lay the foundations for more precise and accurate understanding of a patient’s unique tumoral characteristics. Better insights into both the quantitative and qualitative aspects of a patient’s disease may help open opportunities for enhanced patient care and outcomes in the future.

## Figures and Tables

**Figure 1 cancers-11-00829-f001:**
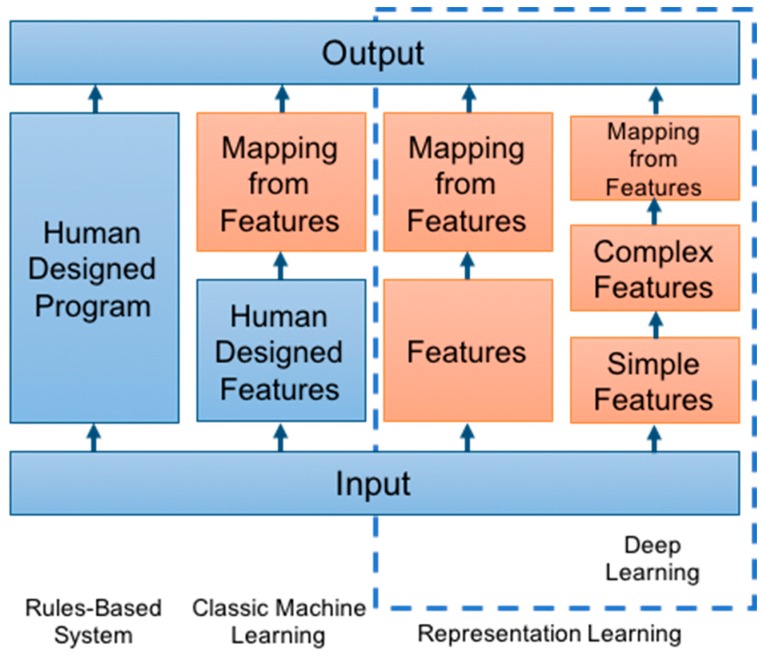
Adapted from Goodfellow et al. [7]. Flowchart of the varying machine learning components across different disciplines, increasing in sophistication from left to right. Orange boxes denote trainable components.

**Figure 2 cancers-11-00829-f002:**
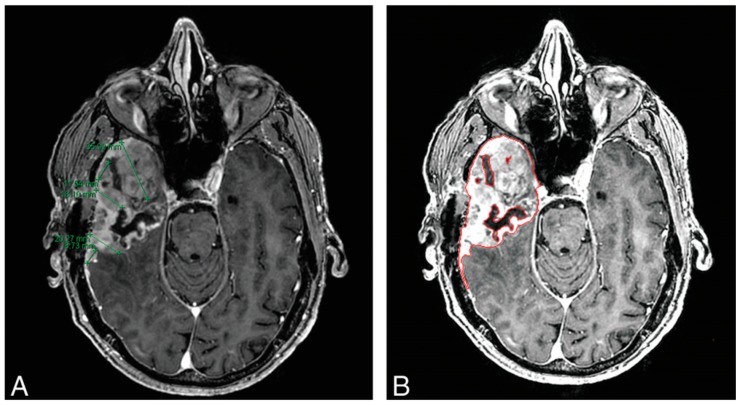
Comparison of linear 1D measurements (**A**) and machine learning volumetric analysis (**B**) in a 64-year-old man with GBM, 11 weeks following resection. Chow et al. [35] found volumetric analysis preferable given the irregularity of recurrence. Panel A indicates the challenges of selecting greatest dimensions in 2D, while panel B shows how a semi-automated volumetric approach can accurately capture greatest dimensions.

**Figure 3 cancers-11-00829-f003:**
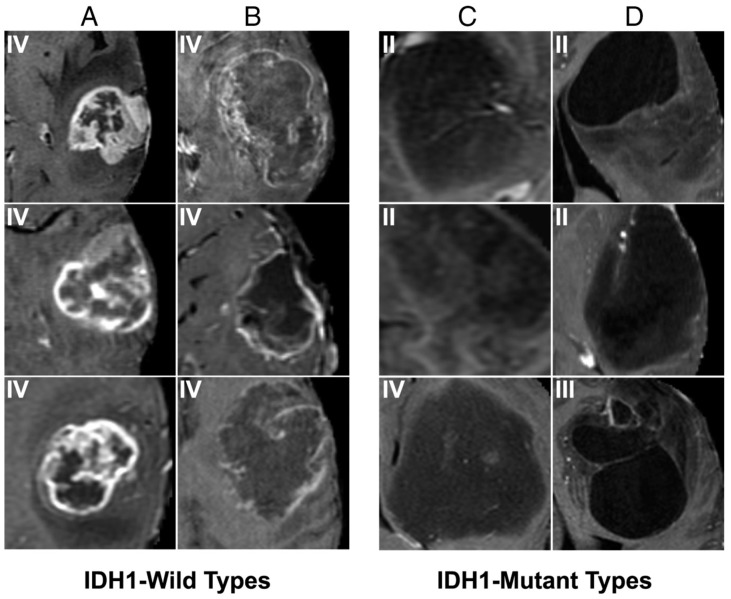
Prototypical imaging features associated with *IDH* mutation status [59]. Our CNN demonstrated that T1 post-contrast features predict *IDH1* mutation status. Specifically, *IDH* wild types are characterized by thick and irregular enhancement (**A**) or thin, irregular, poorly-margined, peripheral enhancement (**B**). In contrast, patients with *IDH* mutations show minimal enhancement (**C**) and well-defined tumor margins (**D**).

**Table 1 cancers-11-00829-t001:** Machine learning architectures and approaches used to segment both pre-operative and post-operative GBMs. Dice score (Sørensen-Dice coefficient) is a statistic used for comparing the spatial overlap of two binary images and is routinely used for tissue classification assessment. Dice scores closer to 1 indicate stronger overlap and accuracy [20].

Author	Approach	Feature	Training Size	Results
Chen et al. [21]	Connected CNN	Necrotic and non-enhancing tumor, peritumoral edema, and GD-enhancing tumor	210 patients	Dice Scores—0.72 whole tumor, 0.81 enhancing tumor, 0.83 core
Havaei et al. [22]	Two Pathway CNN	Local and global features	65 patients	Dice Scores—0.81 whole tumor, 0.58 enhancing tumor, 0.72 core
Yi et al. [23]	3D CNN	Tumor edges	274 patients	Dice Scores—0.89 whole tumor, 0.80 enhancing tumor, 0.76 core
Rao et al. [24]	CNN	Non-tumor, necrosis, edema, non-enhancing tumor, enhancing tumor	10 patients	Accuracy—67%

**Table 2 cancers-11-00829-t002:** A summary of deep learning methods in the characterization of gliomas on MR imaging.

Deep Learning Methods for Glioma Characterization
Author	Year	Character Assessed	Type of DL	Patient Number	MRI Number	Accuracy	AUC	AUPRC	F-1
Bum-Sup Jang et al. [44]	2018	Pseudoprogression	Hybrid deep and machine learning CNN-LSTM	78			0.83	0.87	0.74
Zeynettin Akkus et al. [66]	2015	1p/19q Co-Deletion	Multi-Scale CNN	159		87.70%			
Panagiotis Korfiatis et al. [56]	2017	MGMT Promoter Methylation Status	ResNet50		155	94.90%			
ResNet36	80.72%			
ResNet18	76.75%			
Ken Chang et al. [58]	2018	IDH mutant status	Residual CNN (ResNet34)		406	82.8% training			
83.6% validation
85.7% testing
Peter Chang et al. [59]	2018	IDH mutant Status	CNN	256		94%			
1p/19q Co-Deletion	92%			
MGMT Promoter Methylation Status	83%			
Sen Liang et al. [60]	2018	IDH Mutant Status	Multimodal 3D DenseNet	167		84.60%	85.70%		
Multimodal 3D DenseNet with Transfer learning	91.40%	94.80%		
Jinhua Yu et al. [61]	2017	IDH Mutant Status	CNN Segmentation	110		0.80			
Lichy Han and Maulik Kamdar [62]	2018	MGMT Promoter Methylation Status	Convolutional Recurrent Neural Network (CRNN)	262	5235	0.62 Testing			
0.67 Validation			
0.97 Training			
Zeju Li et al. [65]	2017	IDH1 Mutation Status	CNN	151		92%			
95% (multi-modal MRI)			
Chenjie Ge et al. [63]	2018	High Grade vs. Low Grade Glioma	2D-CNN	285	285	91.93% Training			
93.25% validation			
90.87% test			
1p/19q Co-Deletion	159	159	97.11 training			
90.91% validation			
89.39% test			
Peter Chang et al. [64]	2017	Heterogeneity/ Cellularity	CNN	39	36 MRI, 91 Biopsies	r = 0.74			
Ilya Levner et al. [55]	2009	MGMT Promoter Methylation Status	ANN	59		87.70%

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
