# Peer review of "Optimizing Neuro-Oncology Imaging: A Review of Deep Learning Approaches for Glioma Imaging"

_cancers, 2019, doi:10.3390/cancers11060829_

Round 1
Reviewer 1 Report
Shaver et al provide a pertinent and timely review of deep learning approaches that have the potential to assist the diagnosis, treatment and prognosis of brain tumors based on radiological images. As the audience for this journal includes scientists and clinicians with no computer science background, and as the information provided is highly relevant for the biomedical community, I suggest the authors describe in more detail, in lay terms, and with the help of diagrams and graphics some of the concepts they mention. Images/graphs for other concepts would also enhance the quality of this review article. Please refer to the points below for further detail.
- Lines 52-54: add an image to exemplify lower levels of contrast and lower levels of non-enhanced tumor
-Line 64: add a diagram or use lay terms to explain what multivariate analysis with decision-tree model is
-Line 67: give an example of what "a priori feature selection" typically is in the context of brain tumors
-Lines 76-79: rephrase and use lay terms, accessible to the biomedical community in general. Alternatively, use a diagram/figure.
-Table 1: introduce what a dice score is (it comes later in the text) and discuss/propose what could be deemed as high or low dice score in the context of radiological image interpretation for clinical use.
-Figure 2: if there is a correlation (or lack of) between the data obtained by machine learning volumetric analysis and accurate prognosis, it would be pertinent to mention it.
-Line 159: introduce first what deep learning is and what traditional machine learning is
-Line 126: start by defining what radiogenomics is and point out that it is purely based on non-labeled radiological imaging (not on conventional genomics).
Author Response
Thank you very much for your response. We have attached a word document to address the comments you have provided.

Reviewer 2 Report
This short review is well written and covers both the most important challenges and promises of the field.
Minor edit:
Lines 80-82 are a repeat of 31-33. This should be either rephrased or removed entirely.
Author Response
Thank you very much for your response. We have attached a word document with responses to your comments.

Reviewer 3 Report
Computer-aided diagnosis using artificial intelligent techniques make tremendous improvement in medical applications. Recently, deep learning, a subtype of machine learning artificial intelligence has gained considerable attention because of promising progression in a wide range of radiology applications. Glioma biology is of complexity resulting in a big challenge in diagnosis, surveillance, prognosis, and decision making. Based on radiographic assessment, a deep learning approach for glioma imaging represents an emerging breakthrough.
In this manuscript, a thorough introduction and summarization of cumulative findings of deep learning on glioma application is interesting. The substantial description on pre- and post-operative tumor segmentation, genetic characterization of tissue, and prognostication is helpful to the readers interested on this topic. The writing and presentation order are easy to comprehensive follow.
The current version is clear and acceptable.
Author Response
Thank you very much for your comments! We have uploaded our response to your comments in the attached word document.

Reviewer 4 Report
Dear authors.
I read this review with a lot of interest and share your belief that it can add substantial value to reader of this journal. It is timely and potentially impactful. However, I must say that I was somewhat disappointed with some details, including the quality of the text in several sections (with the Radiogenomics section being the most significant exception), missed opportunities to provide insightful expert interpretation of the available literature and overall lack of strong, detailed content in some of the sections. Some sections are vague, others make claims that are not correctly substantiated and others seem to be a collection of statements derived from the literature without expert authority. I believe there is significant room for improvement and tried to list examples below.
Abstract
- 25: please clarify the statement regarding the percentage of gliomas / malignant brain tumors. The number quoted should consider the contribution of metastases or be rephrased to primary malignant brain tumors.
- 27: recommend modifying sentence construction to avoid the repeated use of “which”.
Introduction:
- 41: same as above, review relative percentage of gliomas to clarify if the numbers refer to all brain tumors or primary brain tumors.
- 43: “From a research perspective, MRI assessment is important for both enrollments for clinical trials and endpoints for monitoring response to therapies”. Recommend rewriting with different choice of words.
- The first paragraph of the introduction is almost a copy of the initial abstract. Recommend rewriting to avoid simple repetition.
- 54: “Human-designed rule-based systems such as VASARI have improved the reproducibility of glioma interpretation” – please provide reference(s) to back this claim.
- 55: recommend replacing “handful” by a less colloquial term.
- 56: “big data” should be capitalized throughout the text.
- 58: ”… and ultimately make a determination given the input variables. – wording of this sentenced can be improved.
- 60: recommend replacing “generally requiring” by “that usually requires”.
- 61: recommend replacing: “for distinguishing” by “to distinguish”.
- 62: The summary regarding the study from Hu et al is confusing to read in my opinion and does not support the previous statement that “Classic machine learning approaches…improve the accuracy of imaging techniques.”.
- 72: In my opinion the explanation of CNNs here fails to make it more easy to understand to radiologists and physicians in general, which should be one of the target audiences for this paper. Unless a significant limitation in text length exists, this is a missed opportunity to explain it more effectively.
- The Abstract and Introduction sections of this paper state very specifically that the current review will “focus on (1) pre- and post-operative tumor segmentation, (2) genetic characterization of tissue, and (3) prognostication. However, the titles for the subsequent sections are “Quantification of Disease Burden, Pseudoprogression, Radiogenomics and Prognostication”. This creates some confusion to the reader. I don’t understand why the descriptor “Tumor Segmentation” was dropped in favor of “Quantification of Disease Burden” and I don’t exactly understand in which subsection “Pseudoprogression” falls into. In my opinion this structure has to be reconsidered.
Quantification of Disease Burden
- 84: Stating that quantitative metrics are “necessary to guide therapy, provide risk stratification, and prognosticate outcome both pre-operatively and post-operatively” is a little bit of an overstatement. I don’t think the field is advanced enough clinically to be described as “necessary” since few if any of these quantitative metrics have withstood prospective clinical scrutiny. Perhaps it would be more appropriate to say that quantitative metrics “have been advocated to improve…”.
- 92: Why is manual segmentation technically challenging ? There are easy tools clinically available in commercial PACS systems, and may not be very efficient, but not necessarily technically challenging.
- 92: The use of “Therefore” here is not appropriate, because the second statement is not a valid conclusion from the first, in reality, the first statement does not necessarily support the second at all.
- 99: This sentence would benefit from rewording.
- 113: same
- 120-121: necrosis is not the main reason why some glioblastomas demonstrate irregular margins, given the fact that necrosis is usually embedded centrally within the enhancing tumor.
- 121: poor construction, grammatical errors and repetition also an issue in this phrase.
- Figure 2 – I don’t quite understand what the reader was trying to measure manually on the left sided image. This measurement technique does not reflect Macdonald/RANO criteria. It is not clear on the right which method was utilized for volumetric measurement (CNN or manual outlining ?). Please specify and reference the CNN method, if that is the case.
- When discussing segmentation, the text underscores the “need for a reliable, reproducible, and accurate tools for surveillance”, however there is no discussion about any direct comparison between CNNs and bidirectional or manual outlining methods in regards to any of these parameters and the true clinical implications for surveillance. Aren’t there any such studies currently available in the literature ?
- 138: choice of wording can be improved: non tumor tissue, enhancing and non enhancing tumor.
- 138: description of the Meier study is overly simplistic and does not add anything substantial to the discussion or to reader’s knowledge other than to partially state what was done.
Pseudoprogression
- 144: “Presently the only accepted standard…. compromise disease-management changes in an aggressive tumor”. This phrases starts by indicating that there is only 1 accepted standard (choice of singular implied by “the”) and then lists different methods such as invasive tissue sampling, short interval imaging and clinical follow up. This is confusing. Also, short interval imaging and clinical follow-up are not distinct methods and are performed together, so it would be preferable to say that the only accepted methods are tissue sampling and short interval clinical follow up with imaging.
- If the performance of machine learning algorithms for pseudoprogresion evaluation are so effective, why is it that they are not being considered for clinical practice ? Are there any limitations ? Is the evidence strong enough to justify pursuing it ? I was left with many questions unanswered after reading this section.
- 168: “Characterizing genetic features of gliomas is important for both prognosis and response to therapy.” Why is “characterizing genetic features of glioma” important for “response to therapy? Do the authors mean that it’s important to “predict response to therapy”?
Radiogenomics
- This is by far the best written section of the paper.
- 217: minor typo “the” instead of “they”
Prognostication
- In contrast, this is one of the sections that needs the most work. It starts ok but becomes quickly confusing and ends with a summary of a literature review devoid of substance.
- 234: poor construction
-237: grammatical error
- 247: 2929 studies ???
Challenges
- The reader is lead to believe that other than technical limitations in the training process machine learning tools have no other limitations or challenges for clinical implementation, which is not true.
Conclusions
- Starts well. The last two sentences merely list applications and should be reviewed for more impact. The use of “will” instead of “have the potential to” or “may” is not totally inappropriate, but somewhat of an overly positive outlook.
Author Response
Thank you very much for your comments. We have attached our response to comments in a word document.

Round 2
Reviewer 4 Report
Thank you for diligently addressing the comments and concerns.